# Scaffold-Free 3-D Cell Sheet Technique Bridges the Gap between 2-D Cell Culture and Animal Models

**DOI:** 10.3390/ijms20194926

**Published:** 2019-10-04

**Authors:** Ayidah Alghuwainem, Alaa T. Alshareeda, Batla Alsowayan

**Affiliations:** Stem Cell & Regenerative Medicine Unit, Cellular Therapy and Cancer Research Department, King Abdullah International Medical Research Center, King Abdulaziz Medical City, Ministry of National Guard Health Affairs, Riyadh 11426, Saudi Arabia; ayidah1992@gmail.com (A.A.); alsowayanba@NGHA.MED.SA (B.A.)

**Keywords:** 3-dimensional (3-D) cell culture, cell sheet technique, scaffold-based technique, solid tumor model

## Abstract

Various tissue engineering techniques have been created in research spanning two centuries, resulting in new opportunities for growing cells in culture and the creation of 3-D tissue-like constructs. These techniques are classified as scaffold-based and scaffold-free techniques. Cell sheet, as a scaffold-free technique, has attracted research interest in the context of drug discovery and tissue repair, because it provides more predictive data for in vivo testing. It is one of the most promising techniques and has the potential to treat degenerative tissues such as heart, kidneys, and liver. In this paper, we argue the advantages of cell sheets as a scaffold-free approach, compared to other techniques, including scaffold-based and scaffold-free techniques such as the classic systemic injection of cell suspension.

## 1. Introduction

Research focusing on the mechanisms governing tissue or organ development in tissue has mainly been conducted with cell culture and animal models. The majority of biological research is conducted using in vitro two-dimensional (2-D) culture models, though the formation of a monolayer could lead to misleading outcomes. The reason for this is that, within their microenvironment in the human body, cells are surrounded by supporting cellular and non-cellular structures, such as the extracellular matrix (ECM) [1]. In regenerative medicine, cell-based assays have been widely used in research and drug discovery to investigate drug activity, metabolism, and toxicity in vivo [1]. Most researchers are still using the 2-D cell culture approach due to its efficiency, simplicity, and low-cost. However, this approach has limitations [1]. Techniques allowing cells to self-assemble into 3-D designs and reproduce cell–cell interactions in vitro provide power and contribute to the advancement of tissue engineering, 3-D bioprinting, cancer biology, and the pharmaceutical industry.

An approach used in tissue engineering is the in vitro growth of target cells in the required 3-D organ or tissue. Methods for 3-D cell culture are classified as scaffold-free or scaffold-based culture models, with the scaffold being produced by organic or synthetic components. For scaffold-based approaches, 3-D cell culture is done by seeding the cells on a scaffold (porous matrix, such as collagen), where the cells attach and colonize [2]. In the scaffold-free approach, 3-D cultures do not rely on solid supports. The limitations of the classic techniques restrict their implementation in specific areas, including immune rejection when using scaffold-based techniques or cell loss in the circulatory system when using the injection of cell suspension. Scientists are attempting to resolve these limitations with a new technology known as the Cell Sheet Method. Because the cell sheet preserves its ECM, it can be transplanted directly into tissue beds or even stacked, constructing 3-D tissue-like structures.

The purpose of this review is to highlight the potential applications and challenges of the most common scaffold-based/-free methods currently used in 3-D tissue culture. In addition, the progress of the cell-sheet, as a scaffold-free technique, is described and compared with scaffold-based/free methods.

## 2. Scaffold-Based Strategies

Scaffold-based approaches have been used in clinical settings to restore several types of damaged tissue, such as bone, cartilage, ligaments, skin, and skeletal muscle defects [3]. Scaffolds could also function as vehicles to control the delivery of therapeutics, including drugs, proteins, and DNA. Scaffolds are 3-D porous structures consisting of biomaterials which support adequate gas exchange, nutrient supply, and growth factors (GFs) [4].

Most of the cells in the body require ECMs to promote tissue regeneration. ECM provides the cells with structural and mechanical support, including rigidity and elasticity [3]. To create scaffolds that mimic the ECM function, several features, for example, scaffold architecture, tissue compatibility (biodegradable), and mechanical properties should be considered [3].

For the scaffolds to function correctly, the mechanical properties of the scaffolds should match the mechanical properties of the host tissue [3]. Bio-scaffolds mimicking endogenous niches have been used extensively in tissue recovery. However, limited intake of nutrients and cells inside the scaffold can contribute to reduced cell survival and proliferation, and the scaffold implantation may require an invasive surgical procedure. To resolve the problem, Luo et al., [5] developed an injectable 3-D porous micro-scaffold, a non-invasive technique, for cell transplantation and tissue regeneration. Compared to current scaffolds, this micro-scaffold promotes in situ bone regeneration in dogs with cranial defects.

## 3. Scaffold-Free Strategy

The scaffold-free method is a “bottom-up” approach, using a single-cell suspension, spheroid cell aggregates, tissue strands, or cell sheets as building blocks. The process depends on the inherent ability of the building blocks to combine and build larger constructs [6]. Due to the high initial cell seeding density, cell proliferation and migration are not decisive factors in the scaffold-free method, in contrast to the scaffold-based technique, reducing the time required for tissue construction. A notable benefit of this strategy is its ability to create complicated tissue and organ architecture through the fabrication of heterogeneous building blocks composed of distinct cells types [6]. However, the definition of a scaffold-free structure remains vague and questionable. The most frequently used definition is the development of living tissue by using only cells, and depending on the cells create the matrix and architecture. Athanasiou et al., [7] stated that “scaffoldless tissue engineering refers to any platform that does not require cell seeding or adherence within an exogenous, 3-D material”. Some cells, when confluent, are capable of producing high amounts of ECM. This feature has been explored to construct scaffold-free and self-producing 3-D matrix structures, using chondrocytes treated with GFs and matured in a mold. The structure produced proteoglycan and collagen, was implanted in a pig articular defect model and was integrated easily [8].

The scaffold-free techniques can always be applied if cells naturally create a tight intercellular connection, strong enough to enable the cell sheet to be harvested. A thermo-responsive surface can be used to collect a multilayer cell sheet, which can also be used for single-layer cell sheets. Other methodologies have been used for cell sheet harvesting, such as polyvinylidene di-fluoride membrane or Ethylene diamine tetraacetic acid, only for cell sheets with a strong cell-cell connection [9].

Therefore, the concept of a scaffold-free technique is characterized by two processes: (1) the self-organization of cells without the use of external forces (such as scaffold-free bioprinting and cell sheet), and (2) self-assembly depending on natural features without any external forces (non-adherent substrates are applied to allow cells to perform all events with minimal interference, such as spheroid creation). These two processes generate biomimetic tissues with the potential for clinical applications [7].

### 3.1. Cell Sheet Technique 

The cell sheet engineering’ strategy, where cultured cells are harvested as intact sheets together with their deposited ECM, has been constructed using temperature-responsive surfaces and can be used in tissue engineering [10]. The use of “thermo-responsive culture dish” allows cell adhesion and detachment by controlling the hydrophobicity of the surface without the need for any proteolytic enzyme [10] (Figure 1).

A temperature-responsive culture dish is created by the irradiation of an electron beam onto poly(N-isopropylacrylamide) (PIPAAm) on a tissue culture polystyrene dish surface (TCPS), leading to the polymerization of the monomer and the subsequent bounding to the dish surface. PIPAAm is a non-toxic nanoscale material, which allows cell adhesion and proliferation. The bio-functional cell sheets are formed, easily collected and implemented directly for tissue repair and regeneration [10]. If the temperature is higher than 32 °C, the sheet surface is hydrophobic, and the PIPAAm dehydrates, and the cells attach to the plate [10,11]. The cell sheet is detached when the temperature of the surface is lower than 32 °C, making the sheet hydrophilic, hydrating the PIPAAm, which enables reversible cell adhesion [11].

The amount of polymer also influences cell adhesion and detachment from the dishes. The polymer density should be within a margin of 0.8–2.2 µg cm^−2^ [11]. However, culture conditions vary depending on the type of cells used, and each cell type should be tested at various densities and temperatures [11].

The cells seeded on the TCPS adhere and proliferate at 5% CO2 and 37 °C, which causes changes to the cell morphology [10]. When the temperature is lowered to 20 °C, cells lose their flattened morphology and float in the medium. The change in cell morphology is due to the absence of the mediated force between the cytoskeletal and tensile stress of the ECM, resulting in a complete detachment of the cells from the substrate [10]. After detachment, the cells recover and retain their structure, function, ECM integration, and adhesion capabilities.

Cell sheet technology has several advantages, notably the direct transplantation of the sheet in host tissue without a scaffold [10]. A cell sheet can also form layers, creating a 3-D structure. Matsuda et al., [10] attempted to produce a functional organ-like structure, instead of a basic layered structure [10]. In vitro micro-pumps and potential in vivo tubular circulatory support instruments have been explored using myocardial sheets [10]. In the last two decades, sheets of different cell types have been used to regenerate many kinds of tissues or organs, such as heart [12,13,14], liver [15,16], and cornea [15,17,18,19,20].

Table 1 compares the application of cell sheet and the classic cell-suspension injection methods in different types of tissues.

### 3.2. Modeling Solid Tumor Using the Cell Sheet Technique

Cancer is a prevalent cause of mortality globally. A study reported that of all the causes of mortality in the United States, 23% is due to cancer or cancer-related complications [23].

The availability of rodent models to investigate solid cancers offers investigators a tool to increase the knowledge of the bio-pathology signaling pathways by evaluating new therapeutic methods and genetic manipulation techniques [24]. However, some unsolved technical challenges complicate the creation of tumor animal models. These include (1) limited transplantation methods, and (2) the low efficiency of the engraftment of transplanted cancer cells.

A significant problem limiting the value of cancer cell lines as an in vitro model for human cancers, is the fact that these cells are cultured in a 2-D culture model, while cancer is a complex heterotypic disease embedded in an organ with its microenvironment, causing the classic 2-D culture to lose some vital features of its in vivo counterparts [25]. In 2014, Suzuki et al., [26] developed cell sheets using different types of human cancer cell lines including colon (HCT-116), pancreatic (Panc-1), liver (Li-7), and gastric (MKN74). Subsequently, the cell sheets were implanted in nude mice and compared with the conventional cell suspension injection method in terms of their ability to form tumors [26]. The results demonstrated that cancer cell sheets formed a larger tumor volume and more stable engraftment than the conventional cell suspension method. In the cell suspension approach, the cells may be lost through the circulatory system and cannot engraft to form a tumor in the target organs. Even if a uniform number of cells are transplanted, cancerous tissue may not always develop. In contrast, cancer cells transplanted as a cancer cell sheet did not extend outward from the target organ in animal models and infiltrated the target organ deeply [26]. Another limitation of using the tumor cell injection approach is the high chance of tissue damage because of the injection.

Currently, patient-derived xenograft (PDX) models are preferred to cell lines [27]. In the PDX models, the tumor is derived from patients and implanted into animals. This model is considered to be a better predictor of cancer cell response, as they maintain the cellular heterogeneity, architecture, and molecular characteristics of the patient tumor [27]. As a result, PDX models are considered as useful models to examine therapeutic responses of new drugs. However, this model has serious limitations as it requires a fresh patient sample, is costly and time-consuming. A comparison of the cell sheet technique and the classic cell suspension injection method in modeling solid tumors in animals are summarized in Table 2.

## 4. Scaffold-Based Bioprinting or Scaffold-Free Bioprinting

Bioprinting is an expanding technique for the fabrication of artificial tissues or organs via the additive manufacturing of living cells in tissue-specific patterns by loading the cells layer by layer. Recent developments in the manufacturing of scaffold materials and biodegradable hydrogels have supported the rapid growth of 3-D bioprinting and its applications. Two primary methods, scaffold-based and scaffold-free procedures, have emerged. Table 3 summarizes a comparison between scaffold-free and scaffold-based bioprinting. The definition of “scaffold-free” is sometimes misleading, as temporary scaffolding can be used. However, the final biological product must be scaffold-free at the time of implantation or evaluation. Ozbolzt et al., classify decellularized matrix components, hydrogels, and micro-carriers as scaffold-based bioprinting [32]. The two current methods for creating scaffold-free tissue-engineered vascular grafts are sheet seeding and decellularization/recellularization [33]. The latter was classified as scaffold-free, as the scaffold used to create the vessel was significantly degradable, delivering a strong tissue which could be decellularized and applied in in vivo studies with or without recellularization.

Printing living cells to construct tissues is much more complicated than printing plastic or metal items. New biocompatible materials and bioprinting methods are now established, and capable of producing scaffold-free structures. Scaffold-free structures require a higher number of cells and are challenging to create due to the absence of structural stiffness. However, they have many benefits compared to their scaffold-based counterparts. The benefits include faster remodeling, ECM deposition, and integration with the host tissue when implanted, the lack of potentially hazardous chemicals, and physical barriers within the tissues [32].

Lack of progress prompted a combination of different methods to develop synergetic value-added plans. Kachouie et al., [34] chose targeted tissue assembly, expecting that the integration of a scaffold-free and scaffold-based strategy will enhance the development of more complicated functional tissues with physiological architecture, appropriate for clinical applications. Ouyang et al., [35] assembled Mesenchymal Stem Cell (MSC) sheets on a demineralized bone matrix using a wrapping method, which resulted in the differentiation of the MSCs to an osteochondral lineage similar to in situ periosteums. Another application of the synergetic method was created by Chen et al. [36]. They applied poly (DL-lactic-co-glycolic acid) (PLGA) meshes on osteogenic sheets of porcine MSCs, which resulted in tube-like constructs.

In general, synergetic strategies can also be applied in bioprinting and robotic assembly, where each element is placed in alignment according to a pre-defined blueprint. This method is discussed in recent publications using the terminology of bio-fabrication [37,38].

## 5. Vascularization in 3-D Tissues

The achievement of tissue engineering construction depends on vascularization. With normal physiology, blood vessels supply tissues with nutrients and oxygen and waste elimination. The vascularization of transplanted tissues is essential to prevent cell necrosis and maintain the diffusion of nutrients [40]. For the diffusion of nutrients to occur, tissue must be within 100–200 µm of a capillary [40]. Many techniques are used to vascularize tissues including co-culturing target cells with endothelial cells, using GFs, transplantation of multi-layers of cell sheets, using decellularized organs as scaffolds, and using 3-D bioreactors [24,41]. The conventional method for neovascularization of 3-D thin engineered tissues is by waiting for the host blood vessels to expand into the transplanted tissues [24]. A recent study showed that co-culturing endothelial cells with mesenchymal precursor cells improve the vascular network of 3-D tissues following implantation [40]. In addition, treating 3-D tissues with GFs, such as vascular endothelial GF (VEGF) and basic fibroblast GF (bFGF) stimulates angiogenesis. However, the addition of GF may have some negative consequences. Treating 3-D tissues with high or low GF doses could induce abnormal vascularization formation [40], and the addition does not solve the diffusion limitation for thick transplanted tissues [22,40]. To solve this problem, Shimizu et al., [42] fabricated a triple layer of myocardial tissue by stacking the cell sheets creating a dense structure with abundant micro-capillaries that enhanced the vascularization of the transplanted tissues. The study showed that transplanting this construct resulted in tissue that pulsates synchronously, but a construction with 4 or 5 layers caused cell necrosis due to the lack of nutrients and oxygen [42]. This strategy lacks natural vascular networks and transition. Decellularized native tissues or organs with complete vessels have been used as cell scaffolds to resolve this issue. These scaffolds mimic the complex vascular structure of the native tissues. For example, Ott et al., [43] used a complete vascular decellularized rat heart as a scaffold for heart regeneration. Cardiomyocytes and endothelial cells were used for recellularization, resulting in a beating heart. This finding highlights the potential of the decellularization method for the fabrication of in vivo 3-D vascular tissue.

In in vitro generation of a 3-D tissue model, the environmental parameters such as pH, oxygen, and temperature cannot be controlled, which affect cell proliferation [44]. Bioreactors have been used to control the delivery of nutrients and a biomimetic stimulus to control cell growth and differentiation, as well as tissue fabrication. Bioreactors stimulate the growth of different cells such as red blood cells, chimeric antigen receptor T-cells, and induced pluripotent stem cells. They also regulate the biological, biochemical, and biophysical signals leading to the production of a large population of adherent and suspension cells [45].

For fabricating 3-D tissue constructs using bioreactors, the cells are either packed or floating in macro-porous carriers or on networks of fibers [44]. Convective flows are used to form large viable grafts, which they combine with sensors measuring the culture conditions. After the formation of mature 3-D tissues, the fabricated tissues can be transplanted to the host tissue [45]. In a recent perfusion bioreactor study, adipose stem cells were seeded on a scaffold for three weeks, implanted into pigs where they survived for up to 6 months and is considered to be successful neovascularization [45]. The problem with tissue engineering bioreactors is that they form large grafts, which may cause poor vascularization and affect cell viability [45]. To overcome this problem in vivo, tissues are implanted in pockets of omentum or muscles flaps, resulting in complete vascularization of the implanted tissue [45]. Stephenson et al., demonstrated that pre-vascularization of the bone grafts in the omentum is more effective than using stem cells or GFs. Using bioreactors result in increased vascularization compared with the original tissue engineering [45].

## 6. Challenges in Scaffold Fabrication

Construction of 3-D tissues still presents challenges, including poor cell density in the constructed tissue contributing to fibrosis, lack of a microcirculation causing cells necrosis, and adverse inflammatory responses that trigger biodegrading of the scaffolds [46]. The unidentified impact of scaffold products or their interaction with living cells is a significant element to consider when designing scaffolds. Scaffolds could be chemical which may change normal signaling and cell behavior. For example, in pharmaceutical research, unidentified chemical interactions between drugs and scaffold components may occur, resulting in inaccurate data. The porous size of scaffolds affects the diffusion of nutrients, oxygen, and waste, while the material surface affects cell adhesion, growth, and functions [47]. Natural materials, such as bovine, porcine, cadaver tissues, and intestinal submucosa, are associated with poor reproducibility, difficulty in mass production, potential risk of unknown material contaminations, and lack of mechanical strength. Another problem with the use of scaffolds is that using a large size scaffold will require supportive sutures to prevent displacement, collagen being a notable exception [47]. The main challenge of the existing 3-D cell culture techniques is the use of biocompatible scaffolds, which requires tedious synthesis and fabrication steps. A porous 3-D scaffold is notoriously challenging to seed with cells because the distribution of the cells tends to be inhomogeneous, especially in large constructs, as well as the initial low seeding density in comparison to the free-scaffold strategy [6].

Scaffold-free techniques were in their infancy in the early 1990s [48], but evolved faster than scaffold-based techniques due to the structural challenges of constructing complicated geometries and comparatively large constructs without rigid support. Instead of using biodegradable scaffolds, researchers developed scaffold-free methods such as a cell sheet, and the sheets are transplanted into the host tissue without sutures, exogenous structures and barriers to cell-cell/cell-matrix interactions. As a result, a scaffold-free construction method is faster, safer and more reliable [49]. However, the high cost and time needed for the production of each patient-derived cell sheet prevented the development of cell-sheet-based regenerative medicine [50] and the thick layer of cell sheets lead to hypoxia due to the lack of oxygen and nutrients [51]. The lack of a physical barrier may have an impact on mechanotransduction since it can start instantly after assembly or transplantation. The concept that scaffold-free constructs can start merging and remodeling more rapidly, and because they are more permeable to small signaling molecules, may be an advantage for in vivo graft survival, essential to the efficiency of surgical implantation. There is a cost, however. A scaffold-based construct requires a lower number of cells, resulting in a slower method since they need to expand. Starting with a large number of cells is favorable to create environments comparable to those in nature, but cells are costly to acquire and maintain [32].

A significant disadvantage is the lack of a complete understanding of all the factors involved in these techniques, though there has been an increase in the literature related to development procedures and cell (re)programming recently. Table 4 displays a comparison between scaffold-based and scaffold-free strategies in tissue engineering.

## 7. Concluding Remarks and Future Perspectives

Tissue engineering is a multidisciplinary and interdisciplinary field employing principles of chemistry, biology and engineering sciences, and comprises bio-artificial implant fostering and/or the development of tissue remodeling to restore, maintain, or improve the function of tissues or organs. It may include scaffolds with cells and appropriate biochemical signals which promote the 3-D new organ or tissue design [24].

The tissue engineering field is still evolving and diversifying. The conventional scaffold-based method is becoming more advanced, and scaffolds are progressively becoming the so-called instructive and engaging ‘living biomaterials’ from temporary supporting structures. Both scaffold-based and scaffold-free techniques have pros and cons, based on the implementation, which may make them more or less suitable for a specific application. However, despite major advancements, especially in scaffold-based methods, the printing of large-scale tissues and organs remains elusive. Increasing knowledge of cellular and molecular mechanisms, in addition to advances in biological and non-biological materials and machinery, models how tissue engineering is accomplished and how to proceed to achieve more realistic and efficient therapies, besides in vivo designs.

Future in vitro and in vivo research should concentrate on the standardization of culture methods and the investigation of relevant mechanisms of action. Incorporating 3-D bioprinting of scaffolds with automated spheroid deposition will deliver more complicated, mechanically anisotropic tissue engineering structures and accurate 3-D positioning of different cell kinds. Importantly, many limitations of scaffold-based bioprinting can be eliminated with the scaffold-free technique. Cell sheet technology may be the future of regenerative medicine, especially when combined with techniques such as micro-scaffolds (microfluidic). Scaffold-free techniques are advancing rapidly, and modern microfluidic techniques allow the biofabrication of a more complex type of tissue spheroid. Using acoustic tweezers, recent technological developments can significantly boost the development of spheroids [52]. Another prospective route is using ECM molecules, enzymes, and GFs to functionalize scaffolds and tissue spheroids. Coating 3-D scaffolds with a bacterial cell wall element can stimulate or regulate macrophage reactions and induce vascularization of the implanted structure [53].

Apart from the ability to functionalize micro-scaffolds with GFs, the modular nature of the technique may be advantageous in terms of vascularization. For example, the modular structure of micro-gels promotes fast penetration in pre-vascular networks [54]. In a co-culture of MSC and endothelial cells, the cells created a CD31-positive network, engulfing and integrating the micro-gels [55]. Subcutaneous grafting of scaffold-free endothelial cell spheroids may increase capillary networks.

Self-assembly offers significant flexibility in the use and control of many kinds of cells. They can be generated in the shape of a spheroid co-culture or can be introduced by mixing micro-scaffolds comprising of separate cells [6].

The incorporation of magnetic nanoparticles creates new possibilities for the manipulation of spheroids and micro-scaffolds in bottom-up tissue assembly [56]. It is envisaged that a synergetic tissue engineering approach will allow rapid in situ tissue assembly, benefiting from the advancement of new materials and the design of new specific forms of micro-scaffolds to encage tissue spheroids. For the immediate future, one of the most realistic applications of this synergistic approach is cartilage and bone tissue engineering using minimally invasive processes.

In conclusion, although the complex structure of a solid organ with many distinct types of cells and a complex microvascular network remains a challenge, the multidisciplinary aspect of scaffold production enables the growth of new strategies in time. Currently, many scaffold-free 3-D bioprinted constructs are tested in clinical trials in many fields, from the cardiovascular system to dentistry. It is reasonable to assume that this progress will continue in future, with new tissue-based therapies and in vitro methods developed, tested and implemented in practice [32].

## Figures and Tables

**Figure 1 ijms-20-04926-f001:**
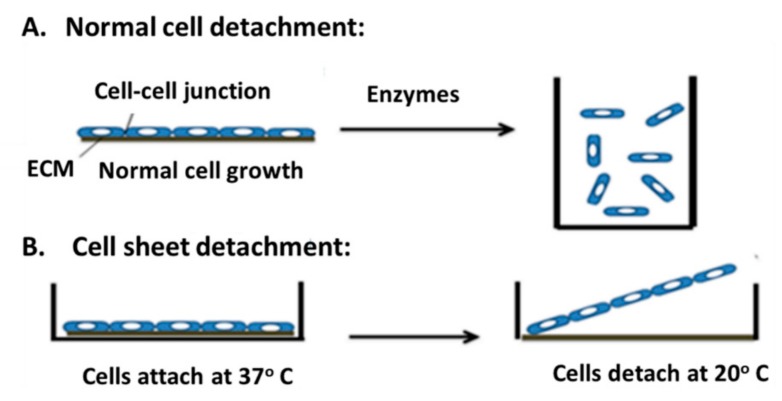
An illustration of standard cell detachment vs cell sheet detachment. (**A**) Cells harvested through enzymatic digestion. In this approach, cells lose their cell–cell junctions which cause them to float in a regular culture dish. (**B**) Cell sheet detachment through temperature control without any enzymatic digestion using temperature responsive culture dish. The temperature is lowered from 37 to 20 °C, which maintains the cell–cell junctions and ECM surface.

**Table 1 ijms-20-04926-t001:** 3-D cell sheet method vs classic cell suspension injection method in clinical and preclinical applications.

Application	Cell Sheet		Direct Classic Injection of Dissociated Cells		Ref.
	Effect	Cell Type	Effect	Cell Type	
**Infarcted Myocardium**	Increases the thickness of left ventricle wall. Decreases the cross sectional left ventricle area. Electrical connections between the implanted cardiomyocyte sheet and the recipient’s heart are developed. Retrieval of damaged cardiac functions. Secretion of several cytokines from the transplanted tissue. Formation of capillary network at the site of myocardial infarction. Remodeling inhibition in the damaged heart. Increased survival and engraftment.	Cardiomyocyte Adipose-derived stromal cell	Poor survival of the cells	BM-MSCs as sources for cardiac muscle cells Adipose-derived stromal cell	[12,13,14]
**Heart Disease**	Improvement in the cardiac function.	Myoblast cell	Hard to control the form, dimensions, or positions of implanted cells.	Skeletal myoblasts	[13,14,21,22]
	1. Improved neovascularization. 2. Enhanced ischemic myocardium function.	Cardiomyocytes + Endothelial cells	Poor survival and engraftment.	Cardiomyocytes + Endothelial cells	
**Ocular Trauma**	1. Patients restore eye vison without any complications. 2. Excessively thin and difficult to handle.	Oral mucosal epithelium sheet	Not tested		[17,18,19,20]
**Retinitis Pigmentosa**	1. The cells migrate into adult retinas. 2. Restores the photoreceptor cells. 3. Differentiate into rod photoreceptors.	Photoreceptor precursor cells	1. Restore Photoreceptors cells 2. Technically more feasible than cell sheet. 3. Restores the photoreceptor cells.	Photoreceptor precursor cells	
**Blindness and Visions**	1. Enhance the eye vison. 2. Improve localized cell delivery and viability. 3. Control of cell density, alignment, and polarity.	Fetal human retina sheets	1. Minimizes surgery time 2. Delay in regeneration 3. Depends on donor neurons’ migratory ability of the to fill empty spots on the degenerating host retina	MSCs	
**Liver Disease**	1. Enhance liver function. 2. Increase rate of survival. 3. Decrease injury level.	Hepatic sheets	1. Weak engraftment of transplanted hepatocytes.	Injecting hepatocytes into spleens or portal veins	[16]
	1. Strength the cell–cell junctions. 2. Enhancing engraftment efficiency. 3. Achieving regulated cell engraftment sites. 4. Produce liver-specific proteins. 5. Limit/prevent anoikis.	iPS-HLC sheets	1. Cells lost their contact with ECMs which resulted in anoikis 2. Difficult to control the engraftment efficiency.	Intrasplenic injection of iPS-HLC	[15]

iPS-HLC = human induced pluripotent stem cell-derived hepatocyte-like cells; BM-MSCs = Bone marrow-Mesenchymal Stem Cells.

**Table 2 ijms-20-04926-t002:** Cell sheet technique vs classic cell suspension injection method in modeling solid tumors in animals.

Target Organ	Cell Sheet		Classic Cell Suspension Injection		Ref.
	Advantages	Disadvantages	Advantages	Disadvantages	
**Any Tumor**	1. Maintain cell–cell junction. 2. A resectable tumor is possible.	1. Technically more complex than the cell suspension injection method. 2. Sometimes it is difficult to handle the sheet (fragile).	1. Fast, cheap and simple. 2. A resectable tumor is possible.	1. Tumor may grow in peripheral sites. 2. Injected cells may be lost in the circulatory system (leakage).	[28]
**Hepatocellular Carcinoma**	1. Successful within 4 weeks. 2. No damage to the liver or surrounding tissues.	1. The surface of the liver is very slippery; therefore, it takes a long time to position the sheet	1. Successful within 4 weeks. 2. Cell preparation takes short time.	1. Difficulty injecting the cells into the organ. 2. Damage of the surrounding tissue.	[29,30,31]
**Colon Cancer**	1. Tumor volume is larger. 2. The cancer cell sheets invaded into the mouse body. 3. Partial invasion of the bones. 4. The tumor did not extend outwards from the transplant site, but infiltrated deeply.	Not discussed	1. Successful within 4 weeks. 2. Cell preparation takes short time.	1. Leakage of the injected cell suspension. 2. The tumors grew subcutaneously and were surrounded by a thin fibrous capsule. 3. The volume of the tumor is smaller.	[26]
**Pancreas Cancer**	1. Tumor volume is larger. 2. The cancer cell sheets invaded into the mouse body. 3. Partial invasion of the bones. 4. The tumor did not extend outwards from the transplant site, but infiltrated deeply.	Not discussed	Not discussed	1. The tumors grew subcutaneously and were surrounded by a thin fibrous capsule. 2. The volume of the tumor is smaller.	[26]
**Gastric Cancer**	1. Tumor volume is larger. 2. The cancer cell sheets invaded into the mouse body. 3. Partial invasion of the bones. 4. The tumors did not extend outwards from the transplant site, but infiltrated deeply.	Not discussed	Not discussed	1. Leakage of the injected cell suspension. 2. The tumors grew subcutaneously and were surrounded by a thin fibrous capsule. 3. The volume of the tumor is smaller.	[26]

**Table 3 ijms-20-04926-t003:** Comparison between scaffold-free bioprinting and scaffold-based bioprinting [39].

Features	Scaffold-Based Bioprinting	Scaffold-Free Bioprinting
Bioprinting Modes	Extrusion, droplet, or laser-based bioprinting	Extrusion-based bioprinting
Resolution and Accuracy	High	Low
Process Time	Short	Medium-long
Bio-Ink	Essentially soft biomaterials;Hydrogel, micro-carriers	Cells produce optimal matrix; Cell pellet, tissue spheroids, and tissue strands.
Intercellular Interaction	Limited communication	Natural interaction/high
Cell Viability	Variable	Higher efficiency
Tissue Bio-Mimicry	Low-medium	High
Preference for Application	Good for large, cell-homogenous, matrix-rich tissue	Best for smaller, cell-heterogeneous, matrix-poor tissues
Affordability	Low-high	High
Commercial Availability	Available	Available

**Table 4 ijms-20-04926-t004:** A comparison between scaffold-based and scaffold-free strategies in tissue engineering.

Feature	Technique Type		
	Scaffold-Free Models		Scaffold-Based Models
	Cell-Sheet	Classic Cell Suspension Injection	
Preparation Time	Fast	Fast	Medium-long
Tissue Regeneration Time	Short	Short	Considerably long
Rate of Cell Survival ^$^	High ^^^	Low	Low
Similarity of Composition to Natural Tissue	Yes	Sometimes ^*^	Sometimes ^**^
Scaffold Degradation	No	Sometimes ^*^	Yes
Proteolytic Enzyme Treatment	No	Yes	Sometimes
Level of Uniform Distribution of the Seeded Cells	High	Low	Low
Immunological Interference from the Scaffold	No	Sometimes ^*^	High
Damage the Surrounded Tissue	No	Possible	Possible ^^^^
Mechanical/Structural Integrity	Fragile and thin sheet	Injectable cells/weak	High
Requirement of a Supportive Sutures	No	No	Yes ^+^
Affordability	High cost	Low cost	Low-high cost
Commercial Availability	Available	Available	Available
Tissue Biomimicry	High	High	Low-medium

^$^ Due to graft site inflammation, mechanical injury and autoimmunity. ^^^ Cells as they do not need to be introduced to many steps, chemicals, compression, temperature, and manipulation. ^*^ Sometimes the cells are mixed with a complex protein mixture (such as Matri-gel). ^**^ Due to the use of synthetic polymers to construct the scaffold. ^^^^ Depend on the type of materials are used to construct the scaffold. ^+^ Require supportive sutures to prevent displacement.

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
