# Peer review of "Scaffold-Free 3-D Cell Sheet Technique Bridges the Gap between 2-D Cell Culture and Animal Models"

_ijms, 2019, doi:10.3390/ijms20194926_

Round 1

Reviewer 1 Report

This review focuses on scaffold-free approaches for regenerative medicine applications. This topic is of interest to the research field and this review could be a significant contribution to the field. However my enthusiasm with this review was affected from the beginning because of the extensive introduction about scaffolds, grammatical errors, modest redaction and some misleading arguments.

In the abstracts the authors start with cancer-related topics; however the main subject of the review is not cancer, and this is misleading at the beginning.

The section describing 2D culture approaches is to lengthily and needs redaction revision.

The fist 4 pages of the review are dedicated to describe the state-of-the-art of regenerative medicine approaches using scaffolds. There are innumerable reviews covering these topics. This section should be shortened.

Some discussion regarding bioprinting techniques compared to scaffold-free approaches should be also considered.

I included annotations on the original pdf. There are so many redaction and grammatical errors that I couldn't comment on everything. I strongly recommend editing help from someone with full professional proofreading in English. 

The authors should focus on reviewing the current state of the art of scaffold-free approaches and minimize the discussion of 2D, 3D, scaffolds, etc.

Author Response

Thank you very much for your comments to improve the submitted work. In this regards, we would like to submit our revised version based on your comments, (below).

1) All the comments, no exceptions, were addressed and the entire review was re-written based on
the comments.
2) The abstract was rewritten, and the sections of 2D and approaches for scaffold were both
shortened.
3) The focus is now on the art of scaffold-free technique as requested.
4) Bioprinter section is added now to the review.
5) The language was edited and revised.

Reviewer 2 Report

The authors present a review on the cell sheet technology, proposing it to have more advantages than the scaffold based technology. Although the topic is interesting, a major revision is suggested in terms of language correction, presentation of the manuscript, and critical analysis of the field. A few comments:

Introduction doesn't give a background of the whole work and why cell sheet technology must be used - rather it talks about regenerative medicine and tissue engineering, and MSCs without connecting it to the cell sheet technology - introduction must be re-written. The last paragraph of the introduction must give an introduce briefly the sections of the paper that is to follow There are various fonts font sizes used in the manuscript which doesn't make a good impression to the reviewers. A "future perspective" section is recommended where the authors' view of the field should be presented.

Author Response

Thank you very much for your comments to improve the submitted work. In this regards, we would like to submit our revised version based on your comments, (below).

1) All the comments, no exceptions, were addressed and the entire review was re-written based on the comments.
2) The introduction was rewritten.
3) The introduction introduces briefly the sections of the paper.
4) The font sizes is corrected
5) A future perspective section is added to the review.
6) The language was edited and revised.

Round 2

Reviewer 1 Report

The authors addressed most of the previous comments and the review has improved compared to the last version.

It still has redaction problems specially in the first half. I recommend the authors to seek professional English proofreading. I dedicated some time to make a few edit suggestions on the original pdf.

The review is focused on scaffold-free techniques, however the space dedicated to review the scaffold-based techniques is to extensive.

In the conclusion the authors introduce the concept of micro-scaffolds, I think this should have been introduced before.

Page numbers are incorrect

Author Response

Manuscript ID: ijms-577491
Thank you very much for your comments to improve the submitted work. In this regards, we would like to submit our revised version based on your comments, (below).
1) I recommend the authors to seek professional English proofreading.
Response:
Professional English proofreading was done. The certificate for the English editing is attached.
2) It still has redaction problems specially in the first half.
Response:
The first half has been revised based on the comment.
3) The review is focused on scaffold-free techniques, however the space dedicated to review the
scaffold-based techniques is to extensive.
Response:
Agree, the review is based on a comparison and filling the gabs with the current scaffold-based
technique. The comment was addressed.
4) In the conclusion the authors introduce the concept of micro-scaffolds, I think this should have
been introduced before.
Response:
The micro-scaffold is under the section of scaffold-based method. Therefore, based on your comment,
this is now mentioned in the first half.
5) Page numbers are incorrect
Response:
Now is corrected.

Thank you for your consideration of this manuscript.

Reviewer 2 Report

All the comments are satisfactorily addressed by the authors.

Author Response

Thanks for you comments.

Round 3

Reviewer 1 Report

The manuscript has been significantly improved. 

Some minor edits:

Line 19: replace systematic with systemic

Line 25: "disease advancement in tissue," not entirely sure what this means, could you rephrase?

The use of "," should be revised throughout the text.

Figure 1 letters are to small

Line 111: 37oC

Author Response

thank you, all the suggested comments have been addressed.